

# Verifiability in computer-aided research: the role of digital scientific notations at the human-computer interface

Konrad Hinsen

Centre de Biophysique Moléculaire (UPR4301), CNRS, Orléans, France
Division Expériences, Synchrotron SOLEIL, Saint Aubin, France

## ABSTRACT

Most of today's scientific research relies on computers and software for processing scientific information. Examples of such computer-aided research are the analysis of experimental data or the simulation of phenomena based on theoretical models. With the rapid increase of computational power, scientific software has integrated more and more complex scientific knowledge in a black-box fashion. As a consequence, its users do not know, and do not even have a chance of finding out, which assumptions and approximations their computations are based on. This black-box nature of scientific software has made the verification of much computer-aided research close to impossible. The present work starts with an analysis of this situation from the point of view of human-computer interaction in scientific research. It identifies the key role of digital scientific notations at the human-computer interface, reviews the most popular ones in use today, and describes a proof-of-concept implementation of Leibniz, a language designed as a verifiable digital scientific notation for models formulated as mathematical equations.

Corresponding author
Konrad Hinsen,
konrad.hinsen@cnrs.fr

# INTRODUCTION

The first few decades of computer-aided research have been marked by the development and application of new computational techniques. They have permitted the exploration of ever more complex systems with ever better precision, but also lead to completely new styles of scientific enquiry based on analyzing large amounts of data by statistical methods. However, the initial enthusiasm about the new possibilities offered by computer-aided research has been dampened in recent years as scientists began to realize that the new technology also brings new kinds of problems. Errors in software, or in the way software is applied, are the most obvious one (*Merali, 2010*; *Soergel, 2014*). A more subtle problem is the widespread non-reproducibility of computational results, in spite of the fact that computations, as defined by Turing in 1937 (*Turing, 1937*), are fully deterministic (*Claerbout & Karrenbach, 1992*; *Stodden et al., 2016*). As a consequence, more and more scientific journals require authors to submit, or publish otherwise, the code and data for computational work described in submitted articles, a recent example being *Nature* (*Nature Editors, 2018*).

But perhaps the most insidious effect of the use of computers is that scientists are losing control over their models and methods, which are increasingly absorbed by software and thereby opacified, to the point of disappearing from scientific discourse (*Hinsen, 2014*). As I will discuss in 'Informal and formal reasoning in scientific discourse', the consequence is that automated computations are much less verifiable than the manually performed computations of the past. Although limits to reproducibility and verifiability have always been a part of scientific research when dealing with observations, these limits are qualitatively and quantitatively different from those introduced by computer-aided research (*Hinsen, 2018*).

In the philosophy of science, these practical questions and more fundamental ones that practitioners do not tend to worry about, are discussed in the context of the *epistemic opacity* of automated computation (*Imbert, 2017*). The overarching issue is that performing a computation by hand, step by step, on concrete data, yields a level of understanding and awareness of potential pitfalls that cannot be achieved by reasoning more abstractly about algorithms. As one moves up the ladder of abstraction from manual computation via writing code from scratch, writing code that relies on libraries, and running code written by others, to having code run by a graduate student, more and more aspects of the computation fade from a researcher's attention. While a certain level of epistemic opacity is inevitable if we want to delegate computations to a machine, there are also many sources of accidental epistemic opacity that can and should be eliminated in order to make scientific results as understandable as possible.

As an example, a major cause for non-reproducibility is the habit of treating an executable computer program, such as the Firefox navigator or the Python interpreter, as an abstraction that is referred to by a name. In reality, what is launched by clicking on the Firefox icon, or by typing "python" on a command line, is a complex assembly of software building blocks, each of which is a snapshot of a continuous line of development. Moreover, a complete computation producing a result shown in a paper typically requires launching many such programs. The complexity of scientific software stacks makes them difficult to document and archive. Moreover, recreating such a software stack identically at a later time is made difficult by the fast pace of change in computing technology and by lack of tool support. A big part of the efforts of the Reproducible Research movement consists of taking a step down on the abstraction ladder. Whereas the individual building blocks of software assemblies, as well as the blueprints for putting them together, were treated as an irrelevant technical detail in the past, this information is now realized as important for reproducibility. In order to make it accessible and exploitable, many support tools for managing software assemblies are currently being developed.

The problem of scientists losing control over their models and methods, leading to the non-verifiability of computations, has a similar root cause as non-reproducibility. Again the fundamental issue is treating a computer program as an abstraction, overlooking the large number of models, methods, and approximations that it implements, and whose suitability for the specific application context of the computation needs to be verified by human experts. To achieve reproducibility, we need to recover control over what software we are running precisely. We must describe our software assemblies in a way that allows

our peers to *use* them on their own computers but also to *inspect* how they were built, for example to check if a bug detected in a building block affects a given published result or not. To achieve verifiability, we need to recover control over which models and methods the software applies. We must describe our model and method assemblies in a way that allows our peers to *apply* them using their own software but also to *inspect* them in order to verify that we made a judicious choice. Reproducibility is about the *technical* decomposition of a computation into software building blocks. Verifiability is about the *scientific* decomposition of a computation into models and methods. As I will show in 'Human-computer interaction in computer-aided research', these two decompositions do not coincide because they are organized according to different criteria.

In this article, I present first results of an ongoing investigation into the causes and mechanisms of non-verifiability in computer-aided research. Contrary to non-reproducibility, which is well understood by now even though effective solutions remain to be developed for many situations, non-verifiability has to the best of my knowledge never been studied so far. My approach combines the scrutiny of today's practices and the design of alternatives that enhance verifiability. I have come to the conclusion that unlike reproducibility, which is best considered an issue of software engineering, verifiability should be treated as an issue of human–computer interaction. In fact, verifiability requires scientists to have a clear notion of the scientific models and methods applied by a piece of software, irrespectively of whether they are performing or reviewing research. The digital scientific notations that are used to encode scientific knowledge at the human–computer interface therefore play an important role.

The contribution made by this work is twofold:

1. An analysis of the obstacles to verifiability in computer-aided research ('Obstacles to Verifiability in Computer-Aided Research')
2. An experimental digital scientific notation designed to support verifiability ('Leibniz, a Digital Scientific Notation for Continuous Mathematics')

The digital scientific notation presented here is a research prototype rather than a tool ready for production use. Its main objective is to show that the obstacles identified in the first part can be overcome in principle. The conclusions ('Conclusions and Future Work') outline the main known limitations, and describe further work required to make verifiable computer-aided research a reality.

## BACKGROUND

### Motivation

The topics I will cover in this article may seem rather abstract and theoretical to many practitioners of computer-aided research. The personal anecdote in this section should provide a more down-to-earth motivation for the analysis that follows. Readers who do not need further motivation can skip this section.

In 1997, I wrote an implementation of the popular AMBER force field for biomolecular simulations (*Cornell et al., 1995*) as part of a Python library that I published later (*Hinsen, 2000*). A force field is a function $U(X, \Phi, G)$ expressing the potential energy of a molecular

system in terms of the positions of the atoms, $X$, a set of parameters, $\Phi$, and a labelled graph $G$ that has the atoms as vertices, the covalent bonds as edges, and an "atom type" label on each vertex that describes both the chemical element of the atom and its chemical environment inside the molecule. Force fields are the main ingredients to the models used in biomolecular simulation, and the subject of much research activity, leading to frequent updates. The computation of a force field involves non-trivial graph traversal algorithms that are habitually not documented, and in fact hardly even mentioned, in the accompanying journal article, which concentrates on describing how the parameter set $\Phi$ was determined and how well the force field reproduces experimental data. I quickly realized that the publication mentioned above plus the publicly available parameter files containing $\Phi$ with their brief documentation were not sufficient to re-implement the AMBER force field, so I started gathering complementary information by doing test calculations with other software implementing AMBER, and by asking questions on various mailing lists.

One of the features of AMBER that I discovered came as a surprise: its potential energy function depends not only on $X$, $\Phi$, and $G$ as defined above, but also on the textual representation of a molecular system in the input files read by a simulation program. Renaming the atom types or changing the order in which the atoms appear in the definition of a molecule can change the potential energy, even though the physical system being described remains exactly the same. I can only speculate about the cause of this design decision, but it is probably the result of simplicity of implementation taking priority over physical reasonableness. A reviewer of the paper would surely have objected had the feature been described there. However, the feature wasn't documented anywhere else than in the source code of a piece of software, which was not published along with the article, and therefore not taken into account during the article's peer reviewing process.

Over the years, I have mentioned this feature to many colleagues, who were all as surprised as I was, and often believed me only after checking for themselves. It has been documented in the meantime (*Field, 2007*, p77), and it has been discussed among software developers (see e.g., *Chodera, Swails & Eastman, 2013*), but for a typical user of molecular simulation software it remains very difficult to discover. It is not obvious either if all software packages implementing AMBER handle this feature in the same way, given that it is in general impossible to obtain identical numbers from different software packages for many other reasons (*Shirts et al., 2017*).

Pragmatists might ask how important this effect is. I do not think anyone can answer this question in general. The numerical impact on a single energy evaluation is very small, but Molecular Dynamics is chaotic, meaning that small differences can be strongly amplified. There are examples of changes assumed to be without effect on the results of MD simulations turning out to be important in the end (e.g., *Reißer et al., 2017*). The hypothesis that AMBER's dependence on unphysical details of the system description has no practical importance would have to be validated for all possible applications of the force field. It would clearly be less effort, at the level of the molecular simulation community, to remove the dependence on unphysical features from the force field definition.

In terms of this specific example, the goal of the work described in this article is to enable the publication of force fields in a form that gives reviewers and users a chance to detect

unphysical features and other potential pitfalls. In order to ensure that optimized simulation software actually uses such a force field as published, it would use the human-readable publication directly as a reference implementation in test suites.

## Verification and validation in science

A prominent feature of scientific research is its elaborate protocol for detecting and correcting mistakes and biases. Its main ingredients are peer verification and continuous validation against new observations. Descriptions of the scientific method tend to emphasize the role of validation as the main error correction technique. It is true that validation alone would in principle be sufficient to detect mistakes. However, the error correction process would be extremely inefficient without the much faster verification steps.

Verification gains in importance as science moves on to more complex systems, for which the total set of observations and models is much larger and coherence of findings is much more difficult to achieve. As an illustration, consider a bug in the implementation of a sequence alignment algorithm in genomics research. Sequence alignment is not directly observable, it is merely a first step in processing raw genomics data in order to draw conclusions that might then be amenable to validation. The path from raw data to the prediction of an observable quantity is so long that finding the cause of a disagreement would be impossible if the individual steps could not be verified.

Peer verification consists of scientists inspecting their colleagues' work with a critical attitude, watching out for mistakes or unjustified conclusions. In today's practice, the first round of critical inspection is peer review of articles submitted to a journal or conference. Most journals do not expect their reviewers to actually re-do experiments or computations, making review a rather shallow process. But peer verification does not stop after publication. If a contribution is judged sufficiently important, it will undergo continued critical inspection by other scientists interested in building on its results.

Verification is possible only if individual contributions are described in sufficient detail that a competent reader can follow the overall reasoning and evaluate the reliability of each piece of evidence that is presented. For computer-aided research, this has become a major challenge. A minimal condition for verifying computations is that the software is available for inspection, as K.V. Roberts called for as early as 1969 in the first issue of the journal *Computer Physics Communication* (*Roberts, 1969*). His advice was not heeded: most scientific software was not published at all, and sometimes even thrown away by its authors at the end of a study. Many widely used software packages were distributed only as executable binaries, with the explicit intention of preventing its users from understanding their inner workings. This development has by now been widely recognized as a mistake and the Reproducible Research movement has been making good progress in establishing best practices to make computations in science inspectable (*Stodden et al., 2016*).

However, the availability of inspectable source code is only the first step in making verification possible. Actually performing this verification is a complicated process in itself, which is often again subdivided into a verification and a validation phase. In the context of software, verification is usually defined as checking that the software conforms to its specification, whereas validation means checking that the specification corresponds to

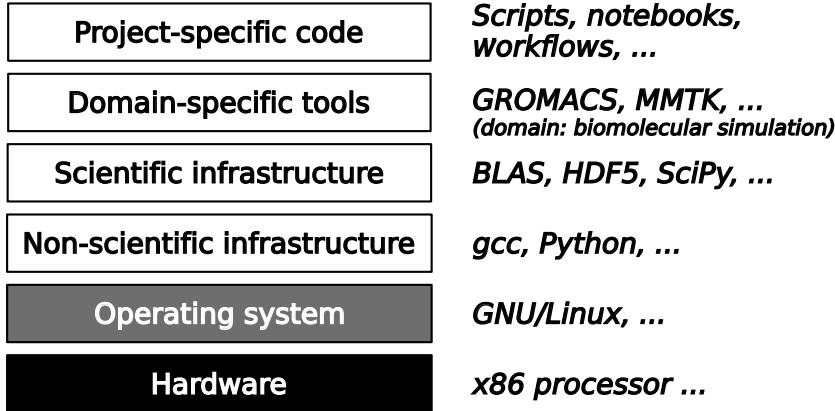

**Figure 1** **A typical software stack in scientific computing consists of fours layers on top of hardware and systems software.** The lower two layers contain widely used infrastructure software that can be verified using generic techniques from software engineering. The upper two layers are specified by scientific discourse and must be verified in its context.

the initial list of requirements. Since the nature of requirements and specifications varies considerably between different domains of application, there is no consensus about the exact borderline between verification and validation. However, as for the scientific method, the general idea is that verification is a faster and more rigorous procedure that focuses on formal aspects, with subsequent validation examining how the software fits into its application context.

Today's practice concerning verification and validation of scientific software varies considerably between scientific disciplines. Well-established models and methods, widely used software packages, and direct economic or societal relevance of results are factors that favor the use of verification and validation. Independently of these domain-specific factors, the place of a piece of software in the full software stack required for a computation determines which verification and validation techniques are available and appropriate. The typical four-layer structure of this software stack is shown in Fig. 1. On a foundation consisting of hardware and systems software, the first layer consists of infrastructure that is not specific to scientific research, such as compilers. From the scientist's point of view, these are commodities provided by the outside world. The next layer consists of scientific infrastructure software that provides widely used algorithms, e.g., for numerical analysis or data management. This generally stable software is developed for research and in contact with scientists, but in terms of verification and validation can be handled like non-scientific infrastructure, because the scientific knowledge embedded into this software consists only of well-known models and methods and the software has a clear though typically informal specification.

The upper two layers are the most difficult ones to verify because there their specifications are incomplete, non-existent, or rapidly evolving. The top layer consists of code written for a specific research project, with the goal of computing something that has never been computed before. It is also unlikely to be reused without modification. This makes

it impossible to apply standard testing procedures. Moreover, quite often this top layer consists of scripts that launch executables written independently and in distinct programming languages, making it difficult to exploit language-centric verification approaches such as static type checking.

One level below, there is a more stable layer of domain-specific tools, which are developed by and for communities of scientists whose sizes range from one to a hundred research groups. In fundamental research, where models and methods evolve rapidly, this domain-specific software is almost as difficult to verify as the project-specific layer. Moreover, it is typically developed by scientists with little or no training in software engineering. For many years, verification and validation of domain-specific tools was very uncommon in fundamental research. Today, widely used community-supported software packages use quality assurance techniques such as unit testing and sometimes continuous integration. However, the scientific validity of the software is still not systematically evaluated, and it is not even clear how that could be achieved. Journals dedicated to the publication of scientific software such as the Journal of Open Source Software (https://joss.theoj.org/; *Smith et al., 2017*) or the Journal of Open Research Software (https://openresearchsoftware.metajnl.com/) do not even ask reviewers to comment on scientific correctness because such a request would be unreasonable given the current state of the art.

## OBSTACLES TO VERIFIABILITY IN COMPUTER-AIDED RESEARCH

The review of the state of the art in the last section shows that verification in computational science is currently focused on software, using software engineering techniques such as testing or formal verification. These techniques compare the behavior of a piece of software to its specification. Verifying computer-aided research also requires the verification of the specification in the context of the scientific question being examined. This raises the question of what that specification actually is, given that clearly identified formal specifications are very rare in most domains of research.

The core of a specification consists of the models and methods that are applied. They are, however, exactly what researchers modify in the course of their work. As a consequence, each computation requires its own *ad-hoc* specification that combines some established models and methods with some more experimental ones into a whole that is usually too complex to be written down in a useful way. The closest approximation to an informal specification is the journal article that describes the scientific work. An essential part of verification is therefore to check if the computation correctly implements the informal description given in the article, or inversely if the journal article correctly describes what the software does. To understand the challenges of this step, it is useful to take a closer look at the interface between scientific discourse and scientific software.

### Informal and formal reasoning in scientific discourse

The main purpose of scientific discourse, whose principal elements today are journal articles and conference presentations, is to communicate new findings in a way that permits peer

**Motion of a mass on a spring**

We consider a point-like object of mass $m$ attached to a spring of force constant $k$ whose mass we assume to be negligible. The other end of the spring is attached to a wall. When the particle is at position $x$, the force acting on it is given by

$$F = -k \cdot d, \tag{1}$$

where $d = x - l$ is the displacement of $x$ relative to the spring's equilibrium length $l$. Newton's equation of motion for the mass takes the form

$$F = m\ddot{x} = -k \cdot (x - l). \tag{2}$$

This second-order ordinary differential equation, which can be rewritten as

$$\ddot{d} = -\frac{k}{m}d \tag{3}$$

in terms of the displacement $d = x - l$, has the solution

$$d(t) = A\cos(\omega t + \delta), \tag{4}$$

where $\omega = k/m$ is the angular frequency of the oscillatory motion, and the amplitude $A$ and phase $\delta$ are arbitrary real numbers.

**Figure 2  Mixing informal and formal reasoning in scientific discourse.** The blue parts describe formal reasoning. The black parts establish the context and define the interpretation of the formal equations.

verification and re-use of the findings in later research. Another category of scientific discourse serves pedagogical purposes: review articles and monographs summarize the state of the art and textbooks teach established scientific knowledge to future generations of scientists. A common aspect of all these writings aimed at experts or future experts is an alternation of informal and formal reasoning. More precisely, formal deductions are *embedded* in an informal narrative.

Before the advent of computers, formal deductions were mainly mathematical derivations and occasional applications of formal logic. In this context, the formal manipulations are performed by the same people who write the informal narratives, with the consequence that the back and forth transitions between the two modes of reasoning, informal and formal, is often blurred. This explains why mathematical notation is often much less formal and precise than its users believe it to be (*Boute, 2005*; *Sussman & Wisdom, 2002*). An illustration is provided in Fig. 2, which shows a simple line of reasoning from elementary physics. Only a careful study of the text reveals that the parts typeset in blue correspond to formal reasoning. One way to identify these parts is to try to replace the textual description as much as possible by output from a computer algebra system. The parts typeset in black introduce the context (Newtonian physics) and define the formal symbols used in the equations in terms of physical concepts.

Computers have vastly broadened the possibilities of formal reasoning through automation. Moreover, the fact that computation enforces a clear distinction of formal and informal reasoning makes it a useful intellectual tool in itself (*Knuth, 1974*; *Sussman & Wisdom, 2002*). However, computing has led to a complete separation of automated

formal reasoning from the informal narratives of scientific discourse. Even in the ideal case of a publication applying today's best practices for reproducible research, the reader has to figure out how text and mathematical formulas in the paper relate to the contents of the various files that make up the executable computation.

This separation creates an important obstacle to verification. In the human-only scenario, both informal and formal reasoning are verified by a single person who, like the author, would not particularly care about the distinction. In the computer-assisted scenario, the narrative on its own cannot be verified because it is incomplete: the formal parts of the reasoning are missing. The computation on its own can be partially checked using software engineering techniques such as testing or static type checking, but in the absence of a specification, verification must remain incomplete. No amount of testing and verifying on the software side can ensure that the computation actually does what it is expected to do. In terms of the illustration of Fig. 2, no verification restricted to the blue parts can establish that Eq. (2) is the correct equation to solve, and no human examination of the black parts can establish that the computation correctly solves Eq. (2).

To the best of my knowledge, neither the frequency nor the causes of mistakes in computer-aided research have ever been the subject of a scientific study. My personal experience from 30 years of research in computational physics and chemistry suggests that roughly half of the mistakes that persist in spite of careful checks at all levels can be described as "the computation was perfectly reasonable but did not correspond to the scientific problem as described in the paper". One typical case is a computation that uses wrong numbers in some place, either wrong constants or incorrectly handled observations. This was the cause for a widely publicized series of retractions of published protein structures, following the discovery of a sign error in data processing software (*Miller, 2006*; *Matthews, 2007*). Another variant is papers describing the computation only incompletely and in such a way that readers assume the computation to be different from what it actually was. A well-known example is the paper by Harvard economists Reinhart and Rogoff that supported austerity politics (*Reinhart & Rogoff, 2010*). It was based on an initially unpublished Excel spreadsheet whose later inspection by independent researchers revealed assumptions not mentioned in the paper, and mistakes in the implementation of some formulas (*Herndon, Ash & Pollin, 2014*). A third variant, perhaps even the most frequent one, is scientists using software without knowing what it does exactly, leaving them unable to judge the well-foundedness of the methods that they are applying. A high-impact example concerning the analysis of fMRI brain scans was recently described (*Eklund, Nichols & Knutsson, 2016*).

It is worth noting that the few well-documented mistakes were found not during reviewing, but only after publication and due to exceptional circumstances or efforts being invested. For example, the incorrect protein structures were detected because they were in contradiction with the structures of similar proteins that were obtained later by different researchers. It was the slow process of validation, rather than routine verification, that brought a software bug into the limelight, a process that incurred a significant cost to the scientific community. Although the small number of proven mistakes in computer-aided research may suggest that they constitute only a minor problem, the real issue is that these

mistakes could not possibly have been detected by the systematic reviewing process that is in principle designed to eliminate them.

It is useful to look at the verification problem as a case of Human–Computer Interaction (HCI), the human part being the informal scientific discourse and the computer part being the computation. The necessity of improving research software through the application of software engineering methods has been pointed out repeatedly (*Naguib & Li, 2010*; *Goble et al., 2016*) and has been translated into concrete recommendations for scientists (e.g., *Taschuk & Wilson, 2017*) and research managers (*Katerbow & Feulner, 2018*). However, all these efforts focus on software. The HCI perspective focuses instead on how software is used by scientists in combination with non-computational reasoning, which is a more constructive point of view for ensuring verifiability.

The popularity of computational notebooks, introduced in 1988 with the computer algebra system Mathematica (*Wolfram Research, Inc., 1988*) and more recently implemented in tools such as Jupyter (*Kluyver et al., 2016*) or R Markdown (*RStudio, Inc., 2016*), shows that the re-unification of informal and formal reasoning into a single document corresponds to a real perceived need in the scientific community. The computational notebook is a variant of the earlier idea of literate programming (*Knuth, 1984*), which differs in that the formal element embedded into the narrative is not a piece of software but a computation with specific inputs and outputs. Its popularity is even more remarkable in view of the restrictions that today's implementations impose: a notebook contains a single linear computation, allowing neither re-usable elements nor an adaptation of the order of presentation to the structure of the surrounding narrative. Both restrictions are a consequence of the underlying computational semantics: the code cells in a notebook are sent, one by one, as input to a command-line interpreter. Non-linear control flow can only happen inside a cell, and there is no way to refer to another cell, or sequence of cells, in order to re-use its contents. Notebooks can therefore capture only the top layer of the software stack shown in Fig. 1, and even that only for relatively simple cases.

Another example for the re-unification of informal and automated formal reasoning is given by textbooks that include short computer programs to explain scientific concepts. Most of them deal specifically with computational science and use the code as examples, e.g., *Langtangen (2012)*, but some aim at conveying more fundamental scientific concepts using executable code for preciseness of notation (*Sussman, Wisdom & Farr, 2013*; *Sussman & Wisdom, 2014*). Unfortunately, the code in such textbooks is in general not directly executable because they were prepared using traditional editing technology in view of being printed on paper. On the other hand, today's computational notebooks are not flexible enough to handle such more complex computational documents, which illustrates that the combination of narratives with computational content is still in its infancy.

## Human–computer interaction in computer-aided research

Computer programs are predominantly viewed as tools, not unlike physical devices such as cars or microscopes. Humans interacting with tools can take on different roles. For scientific software, the main roles are "developer" and "user", whereas for physical devices the number of roles tends to be larger: "designer", "producer", "maintenance

provider", and "user". Software developers interact with software at the source code level using software development tools. Software users interact with the same software via a user interface designed by its developers. Some software provides several levels of user interfaces, for example by providing a scripting or extension programming language for an intermediate category of "power users". Each role in interacting with software is associated with a mental model of how the software works, but these models differ in detail and accuracy. The basic user's mental model is mainly concerned with *what* the software does. A power user knows *how* the software accomplishes its tasks, i.e., what the basic algorithms and data structures are. Developers also need to be aware of the software's architecture and of many implementation details.

In the development of scientific software, these different roles and the associated mental models have so far hardly been taken into account. In fact, in many scientific disciplines there are no clearly defined roles yet that people could adopt. More generally, human–computer interaction in computer-aided research has been shaped by historical accidents rather than by design. In particular, most software user interfaces are the result of a policy giving highest priority to rapid development and thus ease of implementation.

The analysis of verification that I have given in 'Verification and validation in science' and 'Informal and formal reasoning in scientific discourse' suggests that a basic user should be able to acquire a mental model of scientific software that includes everything that may affect the results of a computation. This is a condition for scientists being able to verify the interface between informal and formal reasoning, i.e., to judge if a computation provides an answer to the scientific question being asked. It is not necessary, however, that every user has such a complete mental model of the software's operation. It is common and accepted in scientific work to rely on a smaller number of domain experts for checking models and methods. The main point is that the scientific expertise can be acquired without having to understand orthogonal issues such as performance or implementation details.

In the rest of this article, I will concentrate on human–computer interaction at the user level, focusing on the interplay between informal and computer-assisted formal reasoning in scientific discourse. The overall goal is to explore how computations can be defined in such a way that human scientists can most easily understand and verify them.

### Case study: simulation of the motion of a mass on a spring

As an example, consider the numerical simulation of the motion of a mass on a spring, the simple model system described in Fig. 2. For this particular system, a simulation is of no practical interest because an analytical solution of the equations of motion is known. However, the discussion in this section applies without major changes to any system obeying classical mechanics, including practically relevant systems such as the motion of the planets in the solar system, for which no analytical solution is known.

The equation of motion

$$\ddot{d} = -\frac{k}{m}d \tag{5}$$

with concrete values for the parameters $k$ and $m$ plus initial values $d(t_0)$ and $\dot{d}(t_0)$ define $d(t)$ for all $t$. This information thus constitutes a complete specification for the point mass'

trajectory. Numerical solutions can be obtained after an approximation step that consists of discretization. For simplicity of presentation, I will use the simple Euler discretization scheme, even though it is *not* a good choice in practice. This scheme approximates a first-order differential equation of the form

$$\frac{dz}{dt} = f(t, z(t)) \tag{6}$$

by the discretized equation

$$z(t+h) = z(t) + hf(t, z(t)), \tag{7}$$

which can be iterated, starting from $t = t_0$, to obtain $z(t)$ for a discrete set of time values $t + nh$ for any natural number $n$. The Euler method can be applied to a second-order equation of motion after transforming it into two coupled first-order equations,

$$\dot{d} = v \tag{8}$$
$$\dot{v} = -\frac{k}{m}d, \tag{9}$$

yielding the discretized equations

$$d(t+h) = d(t) + hv(t) \tag{10}$$
$$v(t+h) = v(t) - \frac{hk}{m}d(t) \tag{11}$$

An exact numerical solution can be obtained using rational arithmetic. However, for performance reasons, rational arithmetic is usually approximated by inexact floating-point arithmetic. This approximation involves two steps:

1. The choice of a floating-point representation with associated rules of arithemtic. The most popular choices are the single- and double-precision binary representations of IEEE standard 754-2008.
2. The choice of the order of the floating-point arithmetic operations, which due to rounding errors do not respect the usual associativity rules for exact arithmetic.

The scientific decomposition of this computation thus consists of five parts, each of which requires a justification or discussion in an informal narrative:

(A) The description of the scientific question by Newton's equations of motion.
(B) The values of the constant parameters $k$ and $m$.
(C) The initial values of the system coordinates and velocities, $d(t_0)$ and $v(t_0)$.
(D) The discretization using the Euler method and the choice of $h$.
(E) The choices concerning the floating-point approximation: precision, rounding mode, order of arithmetic operations.

The technical decomposition of a typical implementation of this computation looks very different:

1. A program that implements an algorithm derived from the equations of motion, using partially specified floating-point arithmetic. This program also reads numerical parameters from a file, and calls a function from an ODE solver library.
2. An ODE solver library implementing the Euler method in partially specified floating-point arithmetic.

3. An input file for the program that provides the numerical values of the parameters $k$ and $m$, the initial values $d(t_0)$ and $v(t_0)$, and the step size $h$.
4. A compiler defining the precise choices for floating-point arithmetic.

The transitions from A to 1, from B/C/D to 3, and from D/E to 1/2/4 require human verification because they represent transitions from informal to formal reasoning. The two approximations that require scientific validation are A $\to$ D (the Euler method) and D $\to$ E (floating-point approximation). In this validation, formal reasoning (running the code) is an important tool. The first validation is in practice done empirically, by varying the step size $h$ and checking for convergence. The many subtleties of this procedure are the subject of numerical analysis. The validity of the floating-point approximation would be straightforward to check if the computation could be done in exact rational arithmetic for comparison. This is unfortunately not possible using the languages and libraries commonly used for numerical work, which either provide no exact rational arithmetic at all or require the implementation steps 1/2 to be modified, introducing an opportunity for introducing mistakes.

## Digital scientific notations

A *digital scientific notation* is a formal language that is part of the user interface of scientific software. This definition includes, but is not limited to, formal languages embedded into informal scientific discourse, as in the case of computational notebooks. Another important category contains the file formats used to store scientific datasets of all kinds. As I have explained in 'Human–computer interaction in computer-aided research', the scientific information to be expressed in terms of digital scientific notations includes everything that is relevant at the user level, i.e., everything that has an impact on the results of a computation. This includes in particular scientific models and computational methods, but also more traditional datasets containing, for example, experimental observations, simulation results, or fitted parameters.

Digital scientific notations differ in two major ways from traditional mathematical notation:

1. They must be able to express algorithms, which take an ever more important role in scientific models and methods.
2. They must be adapted to the much larger size and much more complex structure of scientific models and data that can be processed with the help of computers.

The first criterion has led to the adoption of general-purpose programming languages as digital scientific notations. Most published computational notebooks, for example, use either Python or R for the computational parts. The main advantage of using popular programming languages in computational documents is good support in terms of tools and libraries. On the other hand, since these are programming rather than specification languages, they force users to worry about many technical details that are irrelevant to scientific knowledge. For example, the Python language has three distinct data types for sequences: lists, tuples, and arrays. There are good technical reasons for having these separate types, but for scientific communication, the distinction between them and the conversions that become necessary are only a burden. Another disadvantage

is that the source code of a programming language reduces all scientific knowledge to algorithms for computing specific results. This process implies a loss of information. For example, Newton's equations of motion contain information that is lost in a floating-point implementation of its discrete approximation because the latter can no longer be used to deduce general properties of exact solutions.

Domain-specific languages (DSLs) are another increasingly popular choice for representing scientific knowledge. In contrast to general-purpose programming languages, DSLs are specifically designed as digital scientific notations, and usually avoid the two main disadvantages of programming languages mentioned above. Most scientific DSLs are embedded in a general-purpose programming language. A few almost arbitrarily selected examples are Liszt, a DSL for mesh definitions in finite-element computation, embedded in Scala (*DeVito et al., 2011*), Kendrick, a DSL for ODE-based models in epidemiology, embedded in Smalltalk (*Bui et al., 2016*), and an unnamed DSL for micromagnetics, embedded in Python (*Beg, Pepper & Fangohr, 2017*). The choice for an embedded DSL is typically motivated by simpler implementation and integration into an existing ecosystem of development tools and libraries. On the other hand, embedded DSLs are almost impossible to re-implement in a different programming language with reasonable effort, which creates a barrier to re-using the scientific knowledge encoded using them. A stand-alone DSL is independent of any programming language, as is illustrated by Modelica (*Fritzson & Engelson, 1998*), a general modeling language for the natural and engineering sciences for which multiple implementations in different languages exist. However, each of these implementations is a rather complex piece of software.

Looking at how scientific DSLs are used in practice, it turns out that both embedded and stand-alone DSLs end up being a user interface for a single software package, or at best a very small number of packages. Adopting an existing DSL for a new piece of software is very difficult. One obstacle is that the existing DSLs can be too restrictive, having been designed for a narrowly defined domain. For embedded DSLs, interfacing the embedding language with the implementation language of the new software can turn out to be a major obstacle. Finally, the complexity of a DSL can be prohibitive, as in the case of Modelica. In all these scenarios, the net result is a balkanization of digital scientific knowledge because for each new piece of software, designing a new DSL is often the choice of least effort.

These considerations lead to two important criteria for good digital scientific notations that existing ones do not satisfy at the same time:

- Generality. While it is unrealistic to expect that a single formal language could be adequate for computer-aided research in all scientific disciplines, it should be usable across narrowly defined domains of research, and be extendable to treat newly discovered scenarios.
- Simplicity. The implementation of user interfaces based on a digital scientific notation should not require a disproportionate effort compared to the implementation of the scientific functionality of a piece of software.

In the next section, I will describe an experimental digital scientific notation that was designed with these criteria in mind, and report on first experiences with simple toy

applications. While it is too early to judge if this particular notation will turn out to be suitable for real-life applications, it illustrates that better digital scientific notations can be designed if their role at the human–computer interface is fully taken into account.

# LEIBNIZ, A DIGITAL SCIENTIFIC NOTATION FOR CONTINUOUS MATHEMATICS

An important foundation of many scientific theories is the mathematics of smoothly varying objects such as the real numbers. This foundation includes in particular geometry, analysis, and linear algebra. In some scientific disciplines, such as physics and chemistry, this is the dominant mathematical foundation. In other disciplines, such as biology, it is one important foundation among others, notably discrete mathematics. The digital scientific notation Leibniz (https://github.com/khinsen/leibniz), named after 17th-century polymath Gottfried Wilhelm Leibniz, focuses on continuous mathematics and its application. Like many computer algebra systems, but unlike common programming languages, it can express functions of real numbers and equations involving such functions, in addition to the discrete approximations using rational or floating-point numbers that are used in numerical work.

The design priorities for Leibniz are:

- Embedding in narratives such as journal articles, textbooks, or software documentation, in order to act as an effective human–computer interface. The code structure is subordinate to the structure of the narrative.
- Generality and simplicity, as discussed in 'Digital scientific notations'.

Before discussing how Leibniz achieves these goals, I will present the language through two illustrative examples.

## Leibniz as a formal mathematical notation

Figures 3–5 show three views of a Leibniz document that is an adaptation of the short text shown in Fig. 2 describing the motion of a mass on a spring. This and other examples are also available online in the Leibniz example collection (http://khinsen.net/leibniz-examples/). Figure 3 shows the author view.

The current proof-of-concept implementation of Leibniz is an extension to the document language Scribble (*Flatt, Barzilay & Findler, 2009*), which is part of the Racket ecosystem (*Felleisen et al., 2015*). This choice was made for convenience of implementation, Scribble being easy to extend in a powerful general-purpose programming language. Leibniz could be implemented on top of any extensible markup language, or as a plug-in for a suitable WYSIWYG-style editor.

Scribble source code is a mixture of plain text and commands, similar to the better known document language LaTeX (*Lamport, 1994*) but much simpler. Commands start with an @ character. Leibniz adds several commands such as @op and @equation, which define elements of Leibniz code. The Leibniz processing tool generates the two other views from the author's input document. Figure 4 shows the reader view, a rendered HTML page, in which the Leibniz code is typeset on a blue background. This makes the transition

```
#lang leibniz

@import["mechanics" "mechanics.xml"]
@import["quantities" "quantities.xml"]

@title{Motion of a mass on a spring}
@author{Konrad Hinsen}

@context["mass-on-a-spring"
         #:use "mechanics/dynamics"
         #:use "quantities/angular-frequency"]{

We consider a point-like object of mass @op{m : M} attached to a
spring whose mass we assume to be negligible. The other end of the
spring is attached to a wall. When the particle is at position
@op{x : T→L}, the force @op{F : T→F} acting on it is proportional
to the displacement @op{d : T→L} of @term{x} relative to the
spring's equilibrium length @op{l : L}:
@inset{
    @equation[def-d]{d = x - l} @linebreak[]
    @equation[force]{F = -(k × d)}
}
where @op{k : force-constant} characterizes the elastic properties
of the spring.

Newton's equation of motion for the position @term{x} of the mass
takes the form
@inset{
    @equation[newton-x]{m × 𝒟(𝒟(x)) = -(k × (x - l))}
}
This is a second-order ordinary differential equation, which can be
rewritten in terms of the displacement @term{d}, yielding
@inset{
    @equation[newton-d]{𝒟(𝒟(d)) = -((k ÷ m) × d)}.
}

Introducing @op{ω : angular-frequency} defined by
@equation{ω = √(k ÷ m)}, the solution can be written as
@inset{
    @equation[solution]{d[t] = A × cos((ω × t) + δ) ∀ t:T},
}
where @op{cos(angle) : ℝ} is the cosine function. The amplitude
@op{A : L} and the phase @op{δ : angle} can take arbitray values.

@smaller{Additional arithmetic definitions for this context:}
@inset{@smaller{
  @op{force-constant × T→L : T→F}
    @linebreak[]
  @op{force-constant ÷ M : angular-frequency-squared}
    @linebreak[]
  @op{√(angular-frequency-squared) : angular-frequency}
    @linebreak[]
  @op{angular-frequency-squared × T→L : T→A}
    @linebreak[]
}}

}
```

**Figure 3** **The author view of a Leibniz document shows Leibniz code embedded in a narrative.** Most of the commands (starting with @) are inherited from the Scribble document language, only @context, @op, @term, and @equation are added by Leibniz.

### Motion of a mass on a spring

by Konrad Hinsen

Context *mass-on-a-spring* uses *mechanics/dynamics* uses *quantities/angular-frequency*

We consider a point-like object of mass $m:M$ attached to a spring whose mass we assume to be negligible. The other end of the spring is attached to a wall. When the particle is at position $x:T{\rightarrow}L$, the force $F:T{\rightarrow}F$ acting on it is proportional to the displacement $d:T{\rightarrow}L$ of $x$ relative to the spring's equilibrium length $l:L$:

**def-d**: $d = x - l$

**force**: $F = -(k \times d)$

where $k:\text{force-constant}$ characterizes the elastic properties of the spring.

Newton's equation of motion for the position $x$ of the mass takes the form

**newton-x**: $m \times \mathscr{D}(\mathscr{D}(x)) = -(k \times (x - l))$

This is a second-order ordinary differential equation, which can be rewritten in terms of the displacement $d$, yielding

**newton-d**: $\mathscr{D}(\mathscr{D}(d)) = -((k \div m) \times d)$.

Introducing $\omega:\text{angular-frequency}$ defined by $\omega = \sqrt{(k \div m)}$, the solution can be written as

**solution**: $d[t] = A \times \cos((\omega \times t) + \delta)$
$\forall\, t : T$,

where $\cos(\text{angle}):\mathbb{R}$ is the cosine function. The amplitude $A:L$ and the phase $\delta:\text{angle}$ can take arbitray values.

Additional arithmetic definitions for this context:

force-constant $\times\ T{\rightarrow}L:T{\rightarrow}F$
force-constant $\div\ M:\text{angular-frequency-squared}$
$\sqrt{\text{(angular-frequency-squared)}}:\text{angular-frequency}$
angular-frequency-squared $\times\ T{\rightarrow}L:T{\rightarrow}A$

---

**Figure 4  The reader view of a Leibniz document.** All code is shown on a blue background.

---

between informal and formal reasoning visible at a glance. The machine-readable view, shown in Fig. 5, is an XML file that represents the code in a very rigid format to facilitate processing by scientific software.

A comparison of Figs. 2 and 4 shows that a scientific document using Leibniz is very similar to a traditional one using mathematical notation. The overall writing style is the same, and the order in which arguments are exposed to the reader is the same as well. One important difference is that every symbol that is introduced to denote some physical quantity must be accompanied by a label defining its *sort*, such as M (mass) or T→L (length as a function of time). Sorts formalize the informal names given to categories of quantities in traditional scientific writing. There are few restrictions on the names of sorts. They can be very short labels such as M, more descriptive ones such as force-constant, or common mathematical symbols such as $\mathbb{R}$ for the real numbers. The only sort definitions that are part of the Leibniz language itself are those concerning integers, rational numbers, and real numbers. The other sorts used in this example are defined in the two Leibniz documents that are imported: quantities and mechanics, both of which can be consulted in the Leibniz example collection (http://khinsen.net/leibniz-examples/) .

Every piece of data in Leibniz has a sort attached to it. The sort of a composite term such as x - l is obtained from the sorts of its sub-terms (x:T→L and l:L) and a corresponding

```xml
1   <leibniz-document>
2     <library>
3       <document-ref id="mechanics">mechanics.xml</document-ref>
4       <document-ref id="quantities">quantities.xml</document-ref>
5     </library>
6     <context id="mass-on-a-spring">
7       <includes>
8         <use>mechanics/dynamics</use>
9         <use>quantities/angular-frequency</use>
10       </includes>
11       <sorts>
12         <sort id="ℝ" />
13         <sort id="angle" />
14         <sort id="T→A" />
15         <sort id="angular-frequency" />
16         <sort id="T→L" />
17         <sort id="angular-frequency-squared" />
18         <sort id="T→F" />
19         <sort id="force-constant" />
20         <sort id="L" />
21         <sort id="M" />
22       </sorts>
23       <subsorts />
24       <vars />
25       <ops>
26         <op id="m">
27           <arity />
28           <sort id="M" />
29         </op>
30         <op id="√">
31           <arity>
32             <sort id="angular-frequency-squared" />
33           </arity>
34           <sort id="angular-frequency" />
35         </op>
36         <op id="ω">
37           <arity />
38           <sort id="angular-frequency" />
39         </op>
40         <op id="_±">
41           <arity>
42             <sort id="force-constant" />
43             <sort id="M" />
44           </arity>
45           <sort id="angular-frequency-squared" />
46         </op>
47         <op id="k">
48           <arity />
49           <sort id="force-constant" />
50         </op>
51         <op id="F">
52           <arity />
53           <sort id="T→F" />
54         </op>
55         <op id="x">
56           <arity />
57           <sort id="T→L" />
58         </op>
59         <op id="_x">
60           <arity>
61             <sort id="angular-frequency-squared" />
62             <sort id="T→L" />
63           </arity>
64           <sort id="T→A" />
65         </op>
66         <op id="cos">
67           <arity>
68             <sort id="angle" />
69           </arity>
70           <sort id="ℝ" />
71         </op>
72         <op id="l">
73           <arity />
74           <sort id="L" />
75         </op>
76         <op id="_x">
77           <arity>
78             <sort id="force-constant" />
79             <sort id="T→L" />
80           </arity>
81           <sort id="T→F" />
82         </op>
83         <op id="δ">
84           <arity />
85           <sort id="angle" />
86         </op>
87         <op id="A">
88           <arity />
89           <sort id="L" />
90         </op>
91         <op id="d">
92           <arity />
93           <sort id="T→L" />
94         </op>
95       </ops>
96       <rules />
97       <assets>
98         <asset id="newton-x">
99           <equation>
100             <vars />
101             <left>
102               <term op="_x">
103                 <term-or-var name="m" />
104                 <term op="𝒟">
105                   <term op="𝒟">
106                     <term-or-var name="x" />
107                   </term>
108                 </term>
109               </term>
110             </left>
111             <condition />
112             <right>
113               <term op="-">
114                 <term op="_x">
115                   <term-or-var name="k" />
116                   <term op="_-">
117                     <term-or-var name="x" />
118                     <term-or-var name="l" />
119                   </term>
120                 </term>
121               </term>
122             </right>
123           </equation>
124         </asset>
125         <asset id="force">
126           <equation>
127             <vars />
128             <left>
129               <term-or-var name="F" />
130             </left>
131             <condition />
132             <right>
133               <term op="-">
134                 <term op="_x">
135                   <term-or-var name="k" />
136                   <term-or-var name="d" />
137                 </term>
138               </term>
139             </right>
140           </equation>
141         </asset>
142         <asset id="newton-d">
143           <equation>
144             <vars />
145             <left>
146               <term op="𝒟">
147                 <term op="𝒟">
148                   <term-or-var name="d" />
149                 </term>
150               </term>
151             </left>
152             <condition />
153             <right>
154               <term op="-">
155                 <term op="_x">
156                   <term op="_±">
157                     <term-or-var name="k" />
158                     <term-or-var name="m" />
159                   </term>
160                   <term-or-var name="d" />
161                 </term>
162               </term>
163             </right>
164           </equation>
165         </asset>
166         <asset id="def-d">
167           <equation>
168             <vars />
169             <left>
170               <term-or-var name="d" />
171             </left>
172             <condition />
173             <right>
174               <term op="_-">
175                 <term-or-var name="x" />
176                 <term-or-var name="l" />
177               </term>
178             </right>
179           </equation>
180         </asset>
181         <asset id="solution">
182           <equation>
183             <vars>
184               <var id="t" sort="T" />
185             </vars>
186             <left>
187               <term op="[]">
188                 <term-or-var name="d" />
189                 <term-or-var name="t" />
190               </term>
191             </left>
192             <condition />
193             <right>
194               <term op="_x">
195                 <term-or-var name="A" />
196                 <term op="cos">
197                   <term op="_+">
198                     <term op="_x">
199                       <term-or-var name="ω" />
200                       <term-or-var name="t" />
201                     </term>
202                     <term-or-var name="δ" />
203                   </term>
204                 </term>
205               </term>
206             </right>
207           </equation>
208         </asset>
209       </assets>
210     </context>
211   </leibniz-document>
```

**Figure 5** **The machine-readable view of the mass-on-a-spring example.**

definition for the subtraction operator. In the case of `x - l`, subtraction is defined in the imported document `quantities`. It contains the declaration `T→L - L : T→L` , which says that the difference of a time-dependent length and a constant length is a time-dependent length. In the absence of this declaration, Leibniz would have considered the term `x - l` erroneous. In an equation, the terms on both sides of the equal sign must have compatible sorts, otherwise Leibniz will signal an error as well. The reader of a Leibniz document can therefore be sure that the definitions have passed a set of consistency and completeness checks. These checks prevent not only many mistakes, but also the ambiguities and inconsistencies that are surprisingly common in traditional mathematical notation, as *Sussman & Wisdom (2002)* and *Boute (2005)* have pointed out.

Terms are made up of operators and subterms. Operators may or may not have arguments. Zero-argument operators stand for objects or quantities, such as `x` or `m`. Operators with arguments, written in round brackets after the operator name, are used much like functions in mathematics or in programming languages. An example is `cos(δ)`. Two-argument operators can also be defined using infix notation, such as `x - l` or `k × d`. Contrary to traditional mathematical notation, Leibniz operators do not obey precedence rules. The order of application must be indicated explicitly using brackets. This explains why a term such as `cos((ω × t) + δ)` requires brackets around the subterm `ω × t`. The sole exception is a chain of identical binary operators at the same level of the expression. For example, `a + b + c` is allowed and equivalent to `(a + b) + c`. This rule has been adopted from the Pyret language (*Pyret Development Team, 2018*) and is a compromise between the familiarity of the precedence rules in mathematics and the ease of not having to remember precedence values for a potentially large number of infix operators.

Leibniz also proposes a small number of special-syntax operators in order to be closer to traditional mathematical notation. The only one used in this example is the square-bracket operator in the term `d[t]`. It is defined in imported document `quantities` as the value of the time-dependent length `d` at time `t`. The other two special-syntax operators are superscript and subscript, the former being used for exponentiation.

Sort and operator declarations can appear in any order inside a context, and may be repeated several times. This facilitates their integration into the embedding prose. For the same reason, an operator declaration implicitly declares the sorts used in it, which helps to avoid redundant sentences that serve no other purpose than to surround an additional sort declaration. In this respect, Leibniz differs significantly from standard programming languages, which treat omission and repetition of declarations as errors. Leibniz' elements are designed to be used as elements of prose.

A distinction that Leibniz enforces whereas it is habitually glossed over in mathematical notation is the one between a quantity having a specific though potentially unknown value and a variable that can take an arbitrary value from a predefined set. In the last equation, labelled **solution**, `t` is a variable that can take any value of sort `T` (time), whereas A, $\omega$ , and $\delta$ have specific values in each particular solution of the equation of motion.

A final point that requires an explanation is the section containing "additional arithmetic definitions" at the end of the example. It contains operator definitions that Leibniz requires for completeness, but which are not important enough for a human reader to be explained

explicitly in the text. In fact, in the eyes of a physicist they may seem superfluous, since they only express well-known relations between physical quantities of various dimensions. Leibniz' sort system is not sophisticated enough to permit an automatic derivation of these declarations from a set of rules defining the principles of dimensional analysis.

The remaining differences between Leibniz and traditional mathematical notation are superficial: equations are identified by labels (typeset in boldface) rather than numbers, and the notation is overall more regular, without any special syntax for common operators such as division or square root.

Up to here I have presented the Leibniz version of the mass-on-a-spring example as an explanation for human readers with added consistency and completeness checks. However, the main benefit of using Leibniz is the possibility to use the machine-readable view shown in Fig. 5 in the user interface of scientific software. For example, an ODE solver could to be told to solve equation `newton-d` for a given set of numerical values for the parameters. Likewise, a computer algebra system could be asked to verify that equation `solution` is indeed the most general solution to `newton-d`. Alternatively, it could itself produce a Leibniz document as output, adding `solution` to a shorter human-written document ending with equation `newton-d`.

The unit of code in Leibniz is called a *context*. A context contains declarations of sorts and operators, as discussed above, and *assets* that build on these declarations. Assets are values identified by a label, such as the five equations in the example, which defines a single context called `mass-on-a-spring`. The precise instructions given to an ODE solver, for example, could be "solve equation `newton-d` from the context `mass-on-a-spring` defined in the file `mass-on-a-spring.xml`".

## Leibniz as an algorithmic language

A prominent feature of today's computational models is that they contain algorithms that cannot be fully described by mathematical equations. The second example, shown in Fig. 6, describes one of the simplest useful numerical algorithms: Heron's method for computing square roots, which is a special case of the Newton–Raphson method for finding roots of polynomials. Starting from an initial estimate for the square root, the algorithm iteratively computes improved estimates until the current one is close enough according to the supplied error tolerance level.

For describing algorithms, Leibniz uses the *term rewriting* approach (*Baader & Nipkow, 1999*) that is also widely used by computer algebra systems because of its similarity to the manual manipulation of mathematical expressions. An algorithm consists of a set of rules that define how input terms are transformed into output terms. Execution of the algorithm consists of applying these rules repeatedly to an input term, until no further rewriting is possible.

A rule consists of a pattern on the left of a double arrow and a replacement term on its right. Whenever a term matches the pattern, it is replaced by the replacement term. Most rules have variables in their pattern and use the same variables in the replacement term. Rules can also specify additional conditions that must hold for the rule to be applied. This is the case for the first rule of Heron's algorithm, which yields the current estimate as the

## Heron's algorithm

by Konrad Hinsen

Heron's algorithm computes the square root of an input number $x$ iteratively, starting from an initial estimate $e$, until the result is correct within a given tolerance $\varepsilon$. It is a special case of Newton's method for finding roots of algebraic equations.

Context *heron*
uses *builtins/real-numbers*

### 1 Heron's algorithm using exact arithmetic

Let heron($x$:ℝnn, $\varepsilon$:ℝp, $e$:ℝnn) : ℝnn be the result of Heron's algorithm for computing the square root of $x$ up to tolerance $\varepsilon$, starting from estimate $e$.

The first step of the algorithm is to check if the current approximation is good enough, in which case it is the final result:

heron($x, \varepsilon, e$) $\Rightarrow e$
  if abs($x - e^2$) $< \varepsilon$

Note that the tolerance applies to $x$ and not to $\sqrt{(x)}$.

Otherwise, a new estimate is computed by taking the average of $e$ and $x \div e$:

heron($x, \varepsilon, e$) $\Rightarrow$ heron($x, \varepsilon, 1/2 \times (e + (x \div e))$)

For convenience, we also allow no initial estimate to be supplied, using a default value of 1:

heron($x$:ℝnn, $\varepsilon$:ℝp) : ℝnn
heron($x, \varepsilon$) $\Rightarrow$ heron($x, \varepsilon, 1$)

The iteration starting from 1 will always converge but could well be inefficient.

**Figure 6** **A Leibniz document describing Heron's algorithm for the computation of a square root.** An extended version that also shows a derived floating-point version of the algorithm is available online (http://khinsen.net/leibniz-examples/examples/heron.html).

final result if it is close enough to the solution. The second rule is a formulation of the iteration step.

Term rewriting is better suited to embedding into prose than the functions or subroutines of traditional programming languages. Heron's method as shown in this example consists of three distinct pieces that can be discussed separately in the surrounding prose: the definition of the function, and the two rules. In contrast, a function definition in a language like C or Python would have to be written as a single code block. Literate programming tools therefore provide their own methods for composing code snippets into valid code blocks, which however make them cumbersome to use.

Like most informal presentations of Heron's method (see e.g., this Wikipedia page), but unlike an implementation in a typical programming language, the algorithm is formulated in terms of real numbers and exact arithmetic. The sort ℝ stands for a real number, ℝnn is the subsort of non-negative real numbers, and ℝp the subsort of positive real numbers. The algorithm can be executed for rational number arguments, as shown in the extended online version (http://khinsen.net/leibniz-examples/examples/heron.html).

Leibniz also includes floating-point number support, including a built-in transformation that converts a context using real numbers, such as heron, into a context that uses floating-point numbers instead. A floating-point version of Heron's algorithm derived in this way is shown in the extended online version (http://khinsen.net/leibniz-examples/examples/heron.html) together with a few test cases. This example also showcases

another user interface feature: in the reader view, computationally derived information is typeset on a green background, making it easy to distinguish from human input typeset on a blue background.

Automatically derived floating-point algorithms could be used at the user interface of scientific software, either as a specification for what to compute, or as a source of reference values in unit tests for a more efficient implementation of the same algorithm. One additional advantage of Leibniz in this situation is that its floating-point operations are unambiguously specified without any risk of compiler optimizations changing the order of operations.

### Leibniz under the hood

The two examples discussed above serve as a showcase for Leibniz's most visible features, but some import design decisions are not apparent in them. In the following, I will briefly describe these decisions, assuming that the reader of this section is somewhat familiar with term rewriting systems.

In terms of computational semantics, Leibniz's main source of inspiration has been the OBJ family of algebraic specification languages (*Goguen et al., 2000*), and in particular its most recent incarnation, Maude (*Clavel et al., 2002*). In fact, the semantics of the current version of Leibniz are a subset of Maude's functional modules, the main missing features being conditional sort membership and the possibility to declare operators as commutative and/or associative.

A Leibniz context is similar to a functional module in Maude. It consists of (1) the definition of an order-sorted term algebra, (2) a list of rewrite rules, and (3) any number of assets, which are arbitrary values (terms, rules, or equations) identified by unique labels. Assets are the main Leibniz feature not present in Maude. They do not enter into the term rewriting process at all, but they are important both in explanations for human readers and at the user interface of scientific software, as shown in the last paragraph of 'Leibniz as a formal mathematical notation'.

A Leibniz document, such as the one shown in Figs. 3–5, is a sequence of such contexts, each of which is identified by a unique name. A context can *use* another context, inheriting its term algebra and its rewrite rules, or *extend* it, in which case it also inherits its variables and assets. In a typical Leibniz document, each context extends the preceding one, adding or specializing scientific concepts. This corresponds to a frequent pattern of informal reasoning in scientific discourse that starts with general concepts and assumptions and then moves on to more specific ones. The "use" relation typically serves for references to contexts imported from other documents that treat a more fundamental theory or methodology. In the example, the context `mass-on-a-spring` uses a context from another document called `mechanics` (http://khinsen.net/leibniz-examples/examples/mechanics.html) that defines the basic physical quantities of point mechanics and their relations. When using or extending contexts, it is possible to specify transformations, in particular for renaming sorts or operators to avoid name clashes. The absence of namespaces in Leibniz is an intentional design decision, reflecting the use of names in traditional mathematical writing.

A small number of builtin contexts defines booleans and a hierarchy of number types with associated arithmetic operations. There is intentionally no standard library of widely applicable contexts for mathematics or physics. In the interest of transparency and understandability, Leibniz authors are encouraged to avoid the creation of large general-purpose documents that inevitably become black-box code. Ideally, the reader of a Leibniz document should be able to read and understand all other documents it relies on with reasonable effort.

A final feature of Leibniz that deserves discussion is its sort system, which serves the same purpose as type systems in programming languages. In this system, directly taken over from Maude, sort and subsort declarations define a directed acyclic sort graph, which in general consists of multiple connected components called *kinds*. Operator declarations assign a sort to each term and a required sort to each argument position of an operator. Mismatches at the kind level, i.e., an argument sort not being in the same connected component as a required sort, lead to a rejection of a term in what resembles static type checking in programming languages. Mismatches inside a kind, however, are tolerated. The resulting term is flagged as potentially erroneous but can be processed normally by rewriting. If in the course of rewriting the argument gets replaced by a value that is a subsort of the required sort, the error flag is removed again. The presence of an error flag on a result of a computation thus resembles a runtime error in a dynamically typed language. This mixed static-dynamic verification system offers many of the benefits of a static type checker, but also allows the formulation of constraints on values that cannot be verified statically. Leibniz uses this feature to define fine-grained subsorts on the number sorts, e.g., "positive real number" or "non-zero rational number".

## Discussion

The three main goals in the development of Leibniz have been (1) its usability as a digital scientific notation embedded in informal narratives, (2) generality in not being restricted to a narrowly defined scientific domain, and (3) simplicity of implementation in scientific software. While the current state of Leibniz, and in particular the small number of test applications that have been tried, do not permit a final judgment on how well these goals were achieved, it is nevertheless instructive to analyze *which* features of Leibniz are favorable to reaching these goals and how Leibniz compares to the earlier digital scientific notations reviewed in 'Digital scientific notations'.

The key feature for embedding is the highly declarative nature of Leibniz. The declarations that define a context and the values that build on them (terms and equations) can be inserted in arbitrary order into the sentences of a narrative. Verification at the informal-formal borderline is as well supported by Leibniz as by traditional mathematical notation. None of the digital scientific notations in use today shares this feature. Order matters only for rewrite rules, which has not appeared to be a limitation in the experiments conducted so far. Leibniz permits to write rules as assets identified by unique labels, and then assemble a list of named assets into a rule set for rewriting, but so far this feature has not found a good use.

Visual highlighting of the formal parts of a narrative (the blue and green background colors) allows readers to spot easily which parts of a narrative can affect a computation. Moreover, the reader can be assured that the internal coherence of all such highlighted information has been verified by the Leibniz authoring tool. For example, an equation typeset on a blue background is guaranteed to use only operators declared in the context and the sorts of all terms have been checked for conformity. In this way, Leibniz actively supports human verification by letting the reader concentrate on the scientific aspects.

Generality is achieved by Leibniz not containing any scientific information and yet encapsulating useful foundations for expressing it. These foundations are term algebras, equational logic, and numbers as built-in terms. This is an important difference in comparison to scientific DSLs. In fact, the analogue of a scientific DSL is not Leibniz, but a set of domain-specific Leibniz contexts. The common foundation makes it possible to combine contexts from different domains, which is difficult with DSLs designed independently. General-purpose programming languages follow the same approach as Leibniz in providing domain-neutral semantic foundations for implementing algorithms. These foundations are usually lambda calculus, algebraic data types, and a handful of built-in basic data types such as numbers and character strings. Leibniz' main advantage in this respect is that equational logic is a more useful foundation for expressing scientific knowledge than lambda calculus.

The principle of factoring out application-independent structure and functionality has a practically successful precedent in data languages such as XML (*Bray et al., 2006*). The foundation of XML is a versatile data structure: a tree whose nodes can have arbitrary labels and textual content. XML defines nothing but the syntax for this data structure, delegating the domain-specific semantics to schemas. The combination of data referring to different schemas is made possible by the XML namespace mechanism. The machine-facing side of Leibniz can be thought of as a layer in between the pure syntax of XML and domain-specific scientific knowledge, providing a semantic foundation for scientific models and methods.

The separation of syntax and semantics in XML is reflected by tools that process information stored in XML-based formats. Domain-specific tools can delegate parsing and a part of validation to generic parsers and schema validators. This same principle is expected to ensure simplicity of implementation for Leibniz. Validating and rewriting terms are generic tasks that can be handled by a domain-independent Leibniz runtime library. Assuming Leibniz is widely adopted, optimized Leibniz runtimes will become as ubiquitous as XML parsers.

Another aspect of Leibniz that facilitates its use in scientific software is the separation of a machine-oriented syntax based on XML from the representations that human users interact with. In contrast, for general-purpose programming languages and stand-alone DSLs, syntax is designed to be an important part of the user interface. This makes it difficult to extract and analyze information stored in a program, because any tool wishing to process the source code must deal with the non-trivial syntax designed for human convenience. Moreover, suitable parsers are rarely available as reusable libraries.

## CONCLUSIONS AND FUTURE WORK

The work reported in this article has focused on the role of digital scientific notations for human–computer interaction, and in particular on the embedding of digital scientific notations in scientific discourse with the goal of facilitating verification by human experts. First experiments with Leibniz have shown that it can be embedded in informal discourse much like traditional mathematical notation. It can therefore be expected that human verification will work in the same way, at least for code that can be structured as a sequence of sufficiently short sections. Only further practice can show if this approach scales to more complex scientific models.

The semantics of the initial version of Leibniz were somewhat arbitrarily chosen to be a subset of Maude, which looked like a good starting point for first experiments. However, these experiments suggest that the language is currently too minimalist for productive use in computer-aided research. In particular, the lack of predefined collections (lists, sets) makes it cumbersome to use for many applications, such as the force fields mentioned as a motivation in 'Motivation'. Another aspect of the language that deserves further attention is the sort system summarized in 'Leibniz as a formal mathematical notation'. Many common value constraints in scientific applications would require value-dependent sorts, in the spirit of dependent types. Examples are the compatibility of units of measure, or of the dimensions of matrices.

The precise role of digital scientific notations such as Leibniz in the ecosystem of scientific software remains to be defined. A theoretically attractive but at this time not very feasible approach would have software tools read specifications from Leibniz documents and perform the corresponding computations. In addition to the fundamental issue that we do not have general automatic methods for turning specifications into efficient implementations, there is the practical issue that today's scientific computing universe is very tool-centric, with users often adapting their research methodology to their tools rather than the inverse. A more realistic short-term scenario sees Leibniz used in the documentation of software packages, which could then contain a mixed informal/formal specification of the software's functionality. This specification could be verified scientifically by human reviewers, and the software could be verified against it using techniques such as testing or formal verification. A scientific study would be documented in another Leibniz document that imports contexts from the software's specification.

Leibniz and digital scientific notations similar to it are also promising candidates for unifying symbolic and numerical computation in science. As the example of the mass on a spring shows, Leibniz can represent not only computations, but also equations. A computer algebra system could process equations formulated in Leibniz, producing results such as analytical solutions, approximations, or numerical solution algorithms, which could all be expressed in Leibniz as well. Corresponding Leibniz contexts could be derived automatically from the OpenMath standard (*OpenMath Society, 2000*).

Finally, there is obviously a lot of room for improvement in the tools used by authors and readers for interacting with Leibniz content. Ideally, the author and reader would work with identical or very similar views, which should be more interactive than plain text or

HTML documents. Much inspiration, and probably also implementation techniques, can be adopted from computational notebooks and other innovations in scientific publishing that are currently under development.

## ACKNOWLEDGEMENTS

I am grateful to Prof. Shriram Krishnamurthi for recommending the adoption of the infix operator rules from the Pyret language.

### Funding

The authors received no funding for this work.

### Competing Interests

The authors declare there are no competing interests.

### Author Contributions

- Konrad Hinsen prepared figures and/or tables, performed the computation work, authored or reviewed drafts of the paper, approved the final draft.

### Data Availability

The code used in preparing this work is available on GitHub: https://github.com/khinsen/leibniz.

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
