# Peer review of "Verifiability in computer-aided research: the role of digital scientific notations at the human-computer interface"

_PeerJ Computer Science, doi:10.7717/peerj-cs.158_

## Round 0.1 · original submission · Major Revisions

I am unsure if major or minor revisions are needed. It depends on how you respond to the comments from the first reviewer. If you do not split the paper, the revisions might be considered minor. If you do spilt, this would certainly be major.

But in either case, I generally agree with the points of the reviewers, and would like to see this paper revised so that it can be published.

·

Basic reporting

The paper is well written, but I have concerns about overall structure and about the lack of evidence to back up some of its statements - please see the general notes below.

Experimental design

Not applicable.

Validity of the findings

Not applicable.

Additional comments

Thank you for giving me the opportunity to review this paper - I enjoyed reading it, and believe that the work is both interesting and valuable, but have three reservations about it in its present form:

1. Even upon second reading, I felt that two papers had been combined to make one: a general discussion of reproducibility (or its lack) in science, and a description of the design of a particular literate programming tool. I recognize that the first motivates the second, but I think the audience for the two halves of the paper is likely different, and recommend splitting.

2. Claims or implications are made throughout the paper about the scale and severity of the reproducibility crisis in scientific computing and the usability (or lack thereof) of various software tools, but these are only backed up by specific instances or anecdotes. For example, the author mentions several cases of papers being found to be in error because of computational mistakes, but we simply do not have any solid (quantitative) understanding of how widespread this is - as I note in comments, fewer than 50 of the 5 million papers published 1990-2000 have, to my knowledge, subsequently been withdrawn because of computational errors, which puts our *proven* error rate at 1 in 100,000. Like the author, I believe that the actual rate is much higher than this, but I have no proof, and am increasingly wary of repeating dire warnings that I cannot substantiate while simultaneously telling people that *their* work should be more reproducible.

3. The second part of the paper (describing the design of Leibniz) uses concepts from programming language analysis and design that will be unfamiliar to most people with backgrounds in other sciences. This is not to say that those ideas should be removed - they are as necessary to understanding the work as a knowledge of partial differential equations is to understanding a paper about fluid mechanics - but in places I felt that having worked on this tool for so long, the author may have lost sight of how much background knowledge the description assumes or requires.

I hope these comments and the others embedded in the PDF are helpful, and I would be happy to discuss them directly with the author if so desired.

Thanks,
Greg Wilson

Reviewer 2 ·

Basic reporting

The basic reporting meets requirements in terms of language, citation, figures and scientific notation.

Experimental design

The experimental design is appropriate, although the introduction and background is fairly lengthy. Methods are described with sufficient detail.

Validity of the findings

See main review.

Additional comments

The article by Hinsen focuses on a needs assessment for understanding how users can make mistakes and have difficulty with reproducibility due to the complexity and black-box nature of scientific software. After a fairly lengthy discussion of issues encountered over the authors career, the paper identifies the key role of digital scientific notations at the human-computer interface. A proof of concept implementation of Leibniz as a language for specifying digital scientific notation from models as math equations is one part of a solution. It, however does not entirely solve problems due to the nature of non-determinism in most implementations, needs for a means to specify and understand constants, accuracy and means of implementing the math equations digitally. However, the overall discussion and issues raised are important and the prototype solution via a purposed domain specific language and documentation/code could be a worthwhile path forward. General adoption of such an approach, given the significantly complexity of long term and evolving community scientific/research codes that are growing, highly optimized, and complicated in terms of workflow is unlikely. The scientific validity (with some clarifications noted) appears sufficient and the article appears suitable to join the scholarly literature.

Page 1, lines 38-39: It is stated that computations are fully deterministic. This is typically not true, especially in parallel, due to differing order of operations. Codes can be made fully deterministic, for example with fixed point precision on GPUs, however most often they are not implemented (when optimized) to be fully deterministic. Moreover, reproducibility does not necessarily require determinism.

Page 2, lines 107-108: To this reviewer’s knowledge, even back in 1997, AMBER energies were not dependent in the order of atoms in the coordinate file since they were mapped back to the residue definitions. It may be possible that different order of atoms in the residue definitions could have led to different energies, but this needs to be clarified, i.e. exactly what input file is the author referring to. The issues with implementing force fields is significantly more complicated than simply atom-ordering and there are clear issues about choice of algorithms to implement that may be tricky to figure out without reading the code. For example, the choice of improper angles assigned in AMBER does depend on the alphabetic ordering of the atom types which means if you change a type from CA to ZA that may change the impropers assigned. Moreover, there are issues today with accuracy and speed where many shortcuts are made in the name of performance that may lead to different forces and ultimately results. Even implementation choices such as the value of pi, or the conversion factor from charge to kcal/mol; implementations of cutoffs with or without buffers and automated pairlist updates; use of SHAKE on all bonds (LINCS in Gromacs) versus SHAKE only on hydrogens. I would further note that AMBER has changed in significant ways since 1997 (> 20 years ago). Some issues with energy comparisons among programs is described in JCAMD 31, 147 (2017).

Page 3, lines 125-126: Small effect on energy but due to chaos small differences can be strongly amplified – Do you have evidence for this? Although CHARMM and AMBER by default do not give equivalent energies and forces (although they can be made to get accuracy on forces to 10**-6 if constants are changed), they can give equivalent results on converged conformational sampling in the limits of sufficient sampling. Additionally, equilibrium properties can be converged in MD simulations starting from vastly different initial conditions given sufficient sampling.

Force field errors can occur when re-implemented quite easily leading to anomalous results (for example see recent work of Hayatshahi et al. on dinucleotides JPCB 121, 451 (2017) who could not reproduce the work of Brown et al. JCTC 11, 2315 (2015)).

Page 4, line 163: What is the evidence that “Peer review tends to be shallow”? Provide citation or consider softening the statement – for example, “Given time constraints on reviewers and the complexity of the modern scientific workflow and software that often may involve very large-scale calculations, I would speculate that in some cases initial peer review can be shallow.” Evidence that few reviewers re-do experiments or computations?

In discussion of software only as binary, software not published, you may want to note that in the 90’s publishing of force field parameters became requirement for publication, yet some commercial force fields (Schrodinger) are not published and are proprietary (not available outside of their codes).

---

## Round 0.2 · Minor Revisions

If you can quickly address the minor comments from the two reviewers, we can move this to acceptance - it's quite close now.

·

Basic reporting

See "general comments".

Experimental design

See "general comments".

Validity of the findings

See "general comments".

Additional comments

Thank you for giving me the opportunity to review the changes to this paper - I am grateful to the author for addressing most of my original comments. I still find the transition from the general discussion of verifiability to the specific solution in Leibniz awkward, but I recognize that this is a case of "I wouldn't have written it that way", which isn't a legitimate criticism. The use of the mass-and-spring example throughout is welcome, as is placing the technical discussion of Leibniz in one section; I feel the AMBER anecdote (Section 2.1) could be shortened, but again, that is stylistic rather than substantive.

Recommendation: the paper should be accepted without further changes (other than replacing "literal programming" with "literate programming" on line 373).

Reviewer 2 ·

Basic reporting

No comment.

Experimental design

The experimental design is appropriate, although the introduction, motivation and background are rather lengthy.

Validity of the findings

The findings appear appropriate.

Additional comments

The revision strongly considered and attempted to address concerns raised in previous review. the revised manuscript is improved, although the intro/motivation/background are a little lengthy.

Line 121 - "and the decision might well have been taken by an inexperienced graduate student" is pure speculation and unnecessary to mention. I would suggest omitting this.

Line 123 - "the feature wasn't documented anywhere else than in the source code of a piece of software, which was never peer reviewed at all." Again, speculation without evidence. Many different people have effectively "peer reviewed" the AMBER software. I would suggest removing the phrase "which was never peer reviewed at all".

Line 126-127 "To the best of my knowledge, no paper and no software documentation mentions this behavior" -- omit. It is true that it is non-trivial to figure this out without reading the AMBER code, but there is documentation out there which can be found on Google.

https://books.google.com/books?id=mPgpMig3tx0C&pg=PA77&lpg=PA77&dq=amber+atom+type+ordering+improper&source=bl&ots=h0EegD4GBZ&sig=KQ2jDNV0e8I6-GpOSAvKOUE9Kaw&hl=en&sa=X&ved=0ahUKEwiK89PYitTbAhUSiIMKHY4aCJ4Q6AEITTAG#v=onepage&q=amber%20atom%20type%20ordering%20improper&f=false

From: A Practical Introduction to the Simulation of Molecular Systems
By Martin J. Field (1999).

Also blog posts on OpenMM: https://github.com/pandegroup/openmm/issues/220

Line 129-131 "In fact, I am not even sure that all software implementing AMBER handles this the same way..." See: "Lessons learned from comparing molecular dynamics engines on the SAMPL5 dataset."
Shirts MR, Klein C, Swails JM, Yin J, Gilson MK, Mobley DL, Case DA, Zhong ED.
J Comput Aided Mol Des. 2017 Jan;31(1):147-161. doi: 10.1007/s10822-016-9977-1.


Like 138 - "It would clearly be less effort for everybody to simply fix the force field definition". The definition is fixed / specific, just poorly documented.

---

## Round 0.3 · accepted · Accept

Thanks for making these few changes quickly.